# Analytical Methodology for the Identification of Critical Zones on the Generation of Solid Waste in Large Urban Areas

**DOI:** 10.3390/ijerph17041196

**Published:** 2020-02-13

**Authors:** Johanna Karina Solano Meza, Javier Rodrigo-Ilarri, Claudia Patricia Romero Hernández, Mª Elena Rodrigo-Clavero

**Affiliations:** 1Department of Environmental Engineering, Santo Tomás University, Bogotá 110231, Colombia; 2Instituto de Ingeniería del Agua y del Medio Ambiente (IIAMA), Universitat Politècnica de València, 46022 València, Spain; jrodrigo@upv.es (J.R.-I.); rhclaudiapatri@hotmail.com (C.P.R.H.); marodcla@gmail.com (M.E.R.-C.)

**Keywords:** solid waste, geographic information systems, solid waste management, minimization, collection, final disposal

## Abstract

One of the main environmental issues to address in large urban areas is the ever-increasing generation of municipal solid waste (MSW) and the need to manage it properly. Despite significant efforts having been made to implement comprehensive solid waste management systems, current management methods often do not provide sustainable alternatives which ensure the reduction of solid waste generation. This paper presents an analytical methodology that employs a combination of geographic information system techniques (GIS) along with statistical and numerical optimization methods to evaluate solid waste generation in large urban areas. The methodology was successfully applied to evaluate MSW generation in different exclusive service areas (ASES) of the city of Bogotá (Colombia). The results of the analysis on the solid waste generation data in each collection area in terms of its socioeconomic level are presented below. These socioeconomic levels are explained by defining different strata in terms of their purchasing power. The results demonstrate the usefulness of these GIS and numerical optimization techniques as a valuable complementary tool to analyze and design efficient and sustainable solid waste management systems.

## 1. Introduction

Population growth and industrial development have brought about a consumption pattern that results in environmental, social and economic problems [1]. From an environmental perspective, this entails a critical and constant concern given the lack of control and inadequate solid waste management in cities. Optimal municipal solid waste management requires the integration of waste collection, treatment and disposal, with an emphasis on sustainable management [1], as well as the adoption of concepts and references associated with a circular economy. Therefore, understanding the behavior associated with the increase in the generation of municipal waste in cities becomes indispensable to be able to determine the type of appropriate technology that is required to meet the needs of the population. These studies should be based on a socioeconomic analysis, the population characteristics and the distribution of the waste production, among other variables that can influence solid waste flows within operational management systems. This last problem continues to grow as, “municipal solid waste has increased in volume all over the world. In developing countries in particular this may cause severe impacts to the environment and public health [2]”. Waste management systems require the development of viable solutions that incorporate tools capable of coming up with efficient alternatives for waste management in cities, given that, “municipal solid waste management in dense urban areas is a challenge for municipalities, especially in developing countries, which commonly have deficient waste management [3]”.

Currently, decision makers regularly need fast and effective tools to determine appropriate alternatives for later comparison in accordance with various preconditions or performance criteria. Therefore, efficient solid waste management requires responsible stewardship that extensively reviews and analyzes the needs and desired development managerial actions [4], which require further or complementary analysis to determine their technical, economic and social feasibility. This is in addition to what has been established and regulated within action plans and future urban development projections, which includes different environmental variables in terms of the goals established for the sustainable development of cities. 

However, developing strategy and implementation alternatives is not easy because different parameters, not only experimental and analytical, influence decision making. “Solid waste management leads to resource loss and limited waste management approaches, nevertheless, there are barriers to understanding and challenges to maintaining sustainable solid waste management practices [5]”. This problem has become serious in urban areas, where its inadequate management causes soil, water and environmental pollution, which can affect public health. Solutions to these problems are generally evaluated using traditional methods that require large amounts of data [6], and in the case of quantifying waste and its related databases, standardized data are not commonly found, making it difficult to apply specific and accurate mathematical models.

Therefore, municipal solid waste management in large cities requires the implementation of strategies, programs and technologies from local administrations that allow for the adequate collection, management and final disposal of waste. This paper introduces a general methodology that could be applied to analyze the solid waste generation in large cities and its application to the case study of the megacity of Bogotá (Colombia). By means of this research, an analytical model is presented. The model determines the increase of solid waste generation over the medium term in the city, while identifying different critical generation zones in order to locate areas that require immediate attention, and thus implement specific actions or technologies that reduce the risk of possible environmental or social damage, in addition to undertaking actions that can be carried out in a timely manner, in accordance with the forecasts generated by the model. Municipal solid waste management is considered a public health service, providing citizens a disposal system for their trash in an environmentally sound and economically viable manner. The quantity and composition of waste generated comprises the basic information necessary for the planning, operation and optimization of waste management systems [7]. In the city of Bogotá, a set of strategies have been implemented for the comprehensive management of municipal waste, which include educating communities, social integration projects, the optimization and improvement of the collection system and designing projects to implement high efficiency technologies. These actions were carried out in order to reduce the amount of solid waste that is disposed at the Doña Juana sanitary landfill, which is close to reaching its total capacity.

The city of Bogotá is divided into 20 administrative districts, each of which has its own mayor and a local administrative council. The city currently has a home waste collection scheme divided into six collection areas (exclusive service areas—ASE). This division allows the public cleaning service to have better coverage. Each ASE groups together a certain number of districts according to their location and provides them with a selective collection scheme for recyclable waste. In addition to receiving household waste from Bogotá, the Doña Juana landfill also receives waste from surrounding municipalities. The program focuses on increasing recycling as well as promoting citizen environmental education campaigns for the proper separation of waste at the source. 

Research studies in this field of knowledge have focused on the sustainable development of urban areas due to population growth, which depletes natural resources in an uncontrollable manner. This field has not only developed new technologies to reduce the usage rate of natural resources, but also strategies to recover waste through proper collection and transport techniques for its management. These approaches employ geographic information systems (GIS) and remote sensing, which prove to be useful tools to support waste management [8]. 

Geographic information systems (GIS) are an important component of this spatial study. When managing urban solid waste, the entities responsible for planning can only make the right decisions if they are well informed about the processes that are carried out in urban areas [9]. This type of tool combined with mathematical models can be used as a reference support to optimize decision-making processes, as they are able to systematically improve the results obtained in different settings or scenarios. The prospect of incorporating GIS into different waste management stages is indispensable, given that it is possible to integrate these systems into an analysis of selective municipal solid waste collection possibilities and their appropriate use, in order to obtain reliable data on management performance and costs [10]. GIS have also been integrated into the storage stage of urban solid waste management by analyzing “three important factors including number of waste stations, maximum allowed walking distance and container capacity devoted to each station having effect on the performance of waste storage service [11],” in this case, integrating geographical information systems and the response surface methodology [11].

It is important to highlight the performance in optimizing the use of geographic information systems to support management and decision making. While GIS are simply a database system that facilitates access to data, mapping and analysis, these systems have multiple optimization capabilities [12], making this tool very useful for any application associated with process management, not only environmental, but also administrative and social.

This research study presents an analytical methodology and its application to estimate solid waste generation and its distribution in the city of Bogotá. Prediction and forecasting models are useful tools and their application provides reliable support for decision-making processes, which for these analyses in this area use variables such as the number of residents, population age, urban life expectancy and total amount of municipal solid waste to predict the quantity of solid waste and its makeup [13].

A detailed study of the data on the solid waste generation in the city will contribute towards identifying strategic areas to implement preventative actions to reduce waste generation, as well as to determine future actions to improve and optimize solid waste management strategies. Accuracy of municipal solid waste generation forecasting plays an important role for future planning and the waste management system in a city. The characteristics of solid waste generated are different and may vary from municipality to municipality, or from country to country, which is why the accuracy of solid waste generation forecasting of this nature has become work of the utmost importance in the modern era, and this forecasting requires precise municipal solid waste data [14].

Therefore, the study of the per capita generation of solid waste in each collection area, integrating geographic information systems, results in a more specific diagnosis of the generation zones in order to identify and analyze their characteristics through a statistical approach. By identifying the areas of the city that require more attention for the implementation of programs, plans and projects, preventative and corrective actions can be designed and prioritized according to the specific characteristics of each area.

GIS have also been used to design collection routes through simulation models, which not only facilitates having optimum routes but it also reduces costs [15]. This is extremely important as solid waste collection in an urban area is complex, given that of the total amount of money spent on the collection, transport and elimination of solid waste, approximately 50%–70% of the entire operation, is spent on managing this process. A small percentage improvement in the collections operation can result in significant savings in the total cost [16]. Likewise, these tools have been used to locate sites for final waste disposal, given that among the different approaches that have been proposed for municipal solid waste management, landfills continue to be the final destination for waste. Regardless of the technology used to treat solid waste, the incorrect location of landfills can generate increased environmental, social and economic costs. Therefore, appropriate methods are needed to identify possible landfills, such as multicriteria GIS that combines two techniques: a weighted linear combination and an ordered weighted average for decision-making on the location of sites [17]. For this work, the use of GIS is combined with a statistical model for the analysis of urban solid waste generation as a short-term analytical tool to establish its behavioral trend in the city, and thus determine effective strategies that lead to possible solutions to this environmental problem.

## 2. Materials and Methods 

The development of this research study began with gathering data related to solid waste generation in the city of Bogotá in 2016. Projection values for 2020 were computed to analyze the behavior of municipal waste generation in the city, through a fourth-order polynomial regression, with R^2^ = 0.99. Solid waste generation data were provided by the Special Administrative Unit of Public Services (UAESP) [18]. Data were aggregated by collection area and their respective operating company, taking into account the administrative division of the city into districts.

According to the information provided by the National Administrative Department of Statistics (DANE), the population of Bogotá in 2016 was 7,980,001 inhabitants, who disposed of approximately 2,252,922.98 tons of solid waste/year at the sanitary landfill [19]. The municipal solid waste collection system divides the 20 districts into six areas (each one being referred as an ASE). 

ArcGIS software was used to analyze the available data. The spatial analysis was based on general maps from the Special Administrative Unit of the Bogotá District Cadastral, as well as statistical data from the National Administrative Department of Statistics (DANE) and the District Department of Planning. The 2005 Census was used for population data, as well as DANE projections for the year 2020. Once all the information was collected, the analysis was carried out by making maps for 2005, 2015 and a 2020 projection, which took into account the following variables: (i) population, (ii) amount of solid waste collected and (iii) socioeconomic stratification of the urban area of the city by district. Furthermore, the results of the annual solid waste generation per capita of each ASE were compared in order to consider the variability of this indicator according to the stratification information of each ASE. “Socioeconomic stratification is a classification of strata of residential properties that use public utilities; it is employed mainly to charge for household public utility services in a differentiated manner by strata, allowing subsidies to be allocated and contributions to be collected in this area. To the extent that it geographically identifies sectors with different socioeconomic characteristics, it also facilitates: guidance for public investment planning; carrying out social programs such as the expansion and improvement of public services and roads, healthcare and sanitation, as well as education and recreation services in the zones in which they are most needed; a levy different property tax rates per stratum and guide land management” [19,20]. Table 1 shows the percentage of households by stratum and district. Table 1 and the subsequent analysis takes into consideration seven socioeconomic strata (Str 0–Str 6). Str 0 corresponds to households with the lowest purchasing power, while Str 6 corresponds to those with the highest purchasing power. This study utilized this classification, as it is commonly used by local Bogotá administration to specify the distribution of several taxes between local inhabitants. 

Once all the data were obtained and their spatial distribution inside the city were determined, a statistical methodology was introduced, taking into account the existence of different socioeconomic strata inside each ASE, to understand the distribution of solid waste generation inside the ASE and how they compare with each other. The information in Table 1 shows the socioeconomic stratum of the districts was compared with the generation of waste in each exclusive service area (ASE) for the statistical analysis [19,20]. Strata 1, 2 and 3 correspond to the lower socioeconomic strata that represent users with fewer resources. Stratum 4 has a medium-high socioeconomic capacity, while Strata 5 and 6 correspond to the highest strata that represent users with greater economic resources.

The proposed methodology to identify critical zones regarding solid waste generation was based on the definition of solid waste generation per capita of each i stratum (PPCstr,i), expressed in kg/person/year. That is, the amount of waste produced by every person as a function of their stratum was included. In this manner, for this study, the solid waste generation per capita of stratum i (PPCstr,i) is assumed to remain constant regardless of the ASE analyzed. “Production per capita, PPC, is defined as the quantity of solid waste generated by inhabitant per day (kg/[inhabitant day]) [21]”. “The methods used to estimate the PPC (number of loads, weight–volume and mass balance) take into account the amount of waste generated per day and the number of inhabitants in the study area [21]”. For this case, the production analysis was carried out considering the socioeconomic stratum (PPCstr,i), given that it is considered to influence both the generation and composition of solid waste. This is a reasonable assumption as no consumption behavior changes (and to that end, waste generation) of the population of the different strata are known to happen with respect to the ASE where they live. Therefore, PPCstr,i can be estimated by minimizing the objective function shown in Equations (1)–(3):(1)$$\mathrm{Obj}\;\mathrm{Function}=\;\overset{\mathrm{ASES}\;\mathrm{num}.}{\underset{j=1}{\sum }}\gamma_{j}\cdot {\mathrm{Err}}_{j}$$
where
(2)$$\gamma_{j}=\frac{{\mathrm{Population}\;\mathrm{of}\;\mathrm{ASE}}_{j}}{\mathrm{Total}\;\mathrm{population}}$$
(3)$${\mathrm{Err}}_{j}=\mathrm{abs}\left({\mathrm{MSW}}_{{\mathrm{measured}}_{\mathrm{ASE}\;j}}-{\mathrm{MSW}}_{{\mathrm{estimated}}_{\mathrm{ASE}\;j}}\right)$$
and MSW_measured_ and MSW_estimated,_ Equation (3), are the total waste productions measured and estimated inside each ASE. 

Once PPCstr,i was calculated for each stratum, the total solid waste generation of each ASE was obtained as shown in Equation (4):(4)$${\mathrm{MSW}}_{{\mathrm{estimated}}_{\mathrm{ASE}\;j}}=\overset{\mathrm{strata}}{\underset{i=0}{\sum }}\left({\mathrm{PPC}}_{\mathrm{strata}\;i}\cdot {\mathrm{Population}\;\mathrm{strata}\;i}_{\mathrm{ASE}\;j}\right)$$

Therefore, the objective function (Equation (1)) can be written in terms of the difference between the measured and estimated solid waste generation of each ASE, weighted by the proportion of the ASE’s population, with respect to the total population. Minimization of the objective function was performed using numerical methods that consider every possible value of each PPCstr,i and the current values of the population and the measured waste production of each ASE. 

## 3. Results

In 2005, a total solid waste generation of 2746 t/day was estimated for the municipality of Bogotá [22,23]. These data were used as the basis to estimate the solid waste generation of each ASE in 2005. Using all the existing information, the location of the geographic zones of Bogotá with the highest solid waste generation per capita were identified. Results show that in 2005, ASE 3 had the highest solid waste generation per capita, standing at 223 kg/person/year. ASE 3 primarily contains Strata 2, 3 and 4. Furthermore, ASE 3 has the highest proportion of Stratum 6 population in Bogotá (Figure 1). According to information provided by waste management companies in 2015, PPC was estimated between 240 and 290 kg/person/year for most zones. However, an unexpected larger value was found for ASE 3.

A detailed comparison of the evolution of PPC in each ASE between 2005 (Figure 1) and 2015 (Figure 2) shows that PPC decreased in ASE 6 but increased in ASE 4. In general, an increase in the total amount of solid waste generated in Bogotá between 2005 and 2015 was identified. 

To estimate the MSW generation for 2020 (Figure 3), population estimations made by DANE and District Planning Secretary [24] were used. The analysis of data shows that the PPC indicator behavior does not significantly change over time and that ASE 3 is the zone with the highest MSW generation in Bogotá. However, a great deal of interesting information was obtained with a model that considers as parameters the evolution of the population, the MSW generation and the economic stratification. Table 2 shows the evolution over time of the MSW generation for each ASE (t/year) and the waste production per capita expressed in kg/person/year) for the years 2005, 2015 and 2020. 

Figure 1, Figure 2 and Figure 3 show the distribution of MSW generation per capita of each ASE, which was obtained with GIS techniques. In comparing the spatial distribution of MSW production in each of the six ASEs from 2005 to 2015, the following preliminary facts emerge: (i) MSW production increases over time at each ASE and (ii) ASE 3 behaves differently in comparison with every other ASE, with unexpectedly high MSW production values. This last fact justifies discarding data from ASE 3 to obtain the PPCstr,i values, which will be explained when introducing the four simulation scenarios described below. Table 2 shows the evolution over time of the MSW generation for each ASE (t/year) and the waste production per capita (kg/person/year) for the years 2005, 2015 and 2020.

Results obtained so far show that the municipal solid waste generation trends are sustained over time. Therefore, specific technical measurements should be designed in order to properly consider the increase in MSW generation and its implications for the solid waste management system, especially with respect to the landfill disposal phase, which is currently the only feasible option for eliminating waste in Bogotá. The proposed methodology used in this study provides a very valuable tool to understand the MSW distribution on each ASE, while taking into account the economic stratification of the population of large cities. Results obtained when applying this methodology to the city of Bogotá are presented below. Table 3 shows the 2015 population distribution by economic stratum of each ASE. The population is specified in this table by economic stratum in each of the solid waste collection areas. The vast majority of the population belongs to stratum 2 (37.6%) and stratum 3 (35.8%).

As stated above, PPCstr,i values are obtained by using numerical methods to minimize Equation (1). Equation (1) yields different solutions, and therefore different PPCstr,i values can be obtained by imposing certain constraints on the optimization problem. These constraints should represent current conditions of the MSW management system and be defined by the modeler, based on realistic assumptions. When analyzing MSW production in Bogotá, the following four scenarios were considered to estimate PPCstr,i: Scenario 1: Calculation of PPCstr,i, taking into account hard constraints in the minimization process. The proposed methodology was applied to the full set of available data to calculate PPCstr,i values by minimizing the objective function. To obtain realistic results in the minimization process, the following two additional (hard) constraints were imposed: (i) the PPCstr,i value must be greater than 200 kg/person/year and (ii) the maximum absolute error of the difference between the estimated MSW generation value and the real value of MSW generated in a specific ASE must be less than 20%.Scenario 2: Calculation of PPCstr,i, taking into account soft constraints in the minimization process. PPCstr,4 values of the results obtained from Scenario 1 were abnormally high in comparison with the results obtained for the other strata. Consequently, Scenario 2 was designed to understand the behavior of PPCstr,i and improve the minimization process. In this scenario, the following additional conditions were imposed to minimize the objective function: (i) the PPCstr,i value must be greater than 200 kg/person/year, (ii) the maximum absolute error of the difference between the estimated MSW generation value and the real value of MSW generated in a specific ASE must be less than 20% and (iii) the maximum absolute error of the difference between the estimated MSW generation value and the real value of MSW generated in ASE 3 may be higher than 20%, if necessary.

Therefore, the constraints considered for Scenario 2 were softer than those in Scenario 1. A larger error for ASE 3 was admitted, while the conditions of the other ASE were the same as those considered in Scenario 1. 

Scenario 3: Calculation of PPCstr,i, while discarding the data from ASE 3. Results obtained in Scenario 2 indicate that the behavior of ASE 3 was not consistent with the behavior of the other ASE. Therefore, Scenario 3 excluded the data from ASE 3. In this instance, the same additional conditions as those considered in Scenario 1 were imposed to minimize the objective function. The following were the hypotheses considered for Scenario 3: (i) the PPCstr,i value must be greater than 200 kg/person/year, (ii) the maximum absolute error of the difference between the estimated MSW generation value and the real value of MSW generated in a specific ASE must be less than 20% and (iii) data from ASE 3 were discarded.Scenario 4: Calculation of PPCstr,i, while discarding the data from ASE 3 and using a block distribution for PPCstr,i. Scenario 4 was designed to explore the results obtained when discarding the data from ASE 3 while also distributing the PPCstr.i values in three blocks. Therefore, there was one PPCstr,i block for Strata 0, 1 and 2; another for Strata 3 and 4; and a third for Strata 5 and 6. In Scenario 4, the same additional conditions as those considered in Scenario 1 were imposed to minimize the objective function. Therefore, the hypotheses for Scenario 4 were as follows: (i) the PPCstr,i value must be greater than 200 kg/person/year, (ii) the maximum absolute error of the difference between the estimated MSW generation value and the real value of MSW generated in a specific ASE must be less than 20%, (iii) PPCstr,i values were distributed in three blocks that group together Strata 0, 1 and 2, Strata 3 and 4, and Strata 5 and 6, and iv) data from ASE 3 were discarded.

Results of the methodology for the municipality of Bogotá are shown below. Table 4 summarizes the results obtained in each scenario while indicating the behavior of each ASE. Table 5 shows the values of the solid waste generation per capita for every strata and every scenario. Finally, Figure 4 shows as a bar diagram the distribution of the values of the solid waste generation per capita for every strata and every scenario.

## 4. Discussion

Forecasting the behavior of waste generation has become a fundamental part of developing and designing different management plans for a city. To be able to forecast this behavior, different tools have been explored, including statistical analysis [25], comparative analysis to identify the representative variables involved in the generation process [26] and techniques such as the computational approach based on the k-means algorithm and self-organizing map (SOM) [27]. With acceptable confidence levels, all these techniques are appropriate for this type of analysis, to generate useful and accurate information so that municipal administrations can make better decisions when developing management plans. Specifically, for the city of Bogotá, efforts have been made to analyze this behavior using different tools such as support vector machines, neural networks and decision trees [28]. This study presents a scenario analysis using a model linked to an objective function to examine the generation of this waste by collection area. Given the city’s characteristics primarily associated with the large amounts of waste generated, in addition to the characterization of the population in the study area, these scenarios provide a better understanding of the results. By including a socioeconomic analysis and identifying critical areas that need to be prioritized in the development of strategies or technology proposals by using tools such as GIS, this objective function forecasts the city’s solid waste generation in a manner that is close to its real behavior. Furthermore, it identifies areas that require greater attention in a given period of time. Similarly, GIS have proven to be useful in the design stages of integrated urban solid waste management, such as in the storage stage [11]. 

The results obtained in this research study are presented in this manner; they are the outcomes of analyzing the following variables which were identified as the most important in this process: the amount of solid waste generated, the socioeconomic stratification and the population that is directly related to per capita production. Results obtained for Scenario 1 are dramatically different than those obtained from Scenarios 2, 3 and 4. For Scenario 1, the highest value of solid waste generation per capita corresponds to stratum 4 (PPCstr,4 = 503 kg/person/year) while for Scenario 2 the highest value of MSW is PPCstr,1 = 312 kg/person/year and PPCstr,4 = 306 kg/person/year. As shown, the PPC value for stratum 4 greatly decreases for Scenarios 2, 3 and 4 in comparison with the estimate made for Scenario 1 (PPCstr,4 decreases from 503 to 246 kg/person/year), while the objective function also decreases from 38,034 t/year (Scenario 1) to 1147 t/year (Scenario 3). 

Results obtained for Scenario 3 are similar to those obtained for Scenario 2, in terms of the maximum values for solid waste generation per capita, which were obtained in both scenarios for stratum 1 (PPCstr,1 = 312 kg/person/year for Scenario 2 and PPCstr,1 = 305 kg/person/year for Scenario 3). However, Scenarios 2 and 3 greatly differ from the minimum PPC values, the shape of the PPC distribution function and the final value of the objective function, which is greater than the order of magnitude for Scenario 3 (Obj. Func. = 12,238 t/year for Scenario 2, while the Obj. Func. = 1147 t/year for Scenario 3). Therefore, results obtained for Scenario 3 are consistent with the data and provide a very accurate estimate of the PPCstr,i values. In this particular case, the value of the objective function is equal to 8285 t/year, while the sum of the absolute errors of the difference between the MSW estimated generation and the MSW real generation is equal to 50,871 t/year. Note that these values are greater than those obtained for Scenario 3 but less than those obtained for Scenarios 1 and 2.

Results obtained for Scenario 4 show that the maximum MSW PPC generation is obtained for the strata located at the center of the distribution (strata 3 and 4), for which, PPCstr,3 and PPCstr,4 = 275 kg/person/year. However, despite providing a good explanation of the MSW generation process, the accuracy of the results obtained in Scenario 4 is lower than that obtained for Scenario 3, in terms of the value of the objective function (Obj. Func. = 1147 t/year for Scenario 3, while the Obj. Func. = 8285 t/year for Scenario 4). As such, this model is also an appropriate tool for making estimates on solid waste generation in the city, as it provides useful information to plan possible strategies and implement technologies that aim to reduce environmental impacts during an estimated period of time.

## 5. Conclusions

This paper presents an analytical methodology that uses a combination of geographic information system techniques (GIS) and analytical techniques to evaluate the municipal solid waste (MSW) generation on large urban areas. The use of GIS techniques facilitates the development of preliminary calculations that evaluate the municipal solid waste generation inside specific areas of a complex urban environment. GIS are very efficient tools for plotting maps of MSW generation with respect to geographical position, and for the analysis of its evolution over time. Once all the data are obtained and their spatial distribution inside the city is determined, an analytical methodology can be applied to understand the distribution of the solid waste generation in the different districts of a large urban area and how they are related, taking into account the existence of the different socioeconomic strata of the population. An analytical methodology to compute the solid waste production per capita of each stratum i (PPCstr,i) is presented. The methodology is based on minimizing the objective function (Equation (1)), which can be written in terms of the population distribution inside the area while considering their socioeconomic purchasing power. In this study, PPCstr,i is assumed to remain constant, regardless of the zone being analyzed, and therefore, PPCstr,i can be estimated by minimizing the objective function, which is defined in terms of the difference between the measured and estimated solid waste generation of each ASE, weighted by the proportion of the population of the ASE, with respect to the total population. 

This methodology was successfully applied to evaluate the MSW generation of the city of Bogotá (Colombia). The application of GIS techniques enabled the development of an evaluation of the municipal solid waste generation of six different districts (ASE) in the city of Bogotá. ArcGIS was successfully used to support the analysis, resulting in MSW generation maps for the period 2005–2020. The results obtained show that the municipal solid waste generation trends are sustained over time, and consequently, the total MSW generation will increase over time. 

Following these results, specific technical measurements should be designed in order to properly consider the increase in MSW generation and its implications for the solid waste management system, especially with respect to the landfill disposal phase, which is currently the only feasible option for eliminating waste in Bogotá. This statistical methodology was also applied to estimate municipal waste generation per capita by socioeconomic stratum. Four different scenarios were analyzed, which considered different hypotheses about the properties of the PPCstr,i. This methodology identified that the data from ASE 3 (included in Scenarios 1 and 2) were not consistent with the data from the other ASE. Therefore, Scenarios 3 and 4 do not consider ASE 3 data, and their final results, in terms of minimizing the estimation errors are far more accurate. Scenario 3 provided the strongest estimates for PPCstr,i, while Scenario 4 (which considered the same municipal waste generation ratio for three strata groups) provides a solid understanding of the general characteristics of the urban area in terms of the PPCstr,1 values. In the case of Bogotá, further research is needed to verify the consistency of the data (especially for ASE 3). Finally, it is important to state that the proposed methodology can be applied to any large urban area to perform an assessment of the current and future MSW generation, without the need of unavailable complex preliminary data. 

## Figures and Tables

**Figure 1 ijerph-17-01196-f001:**
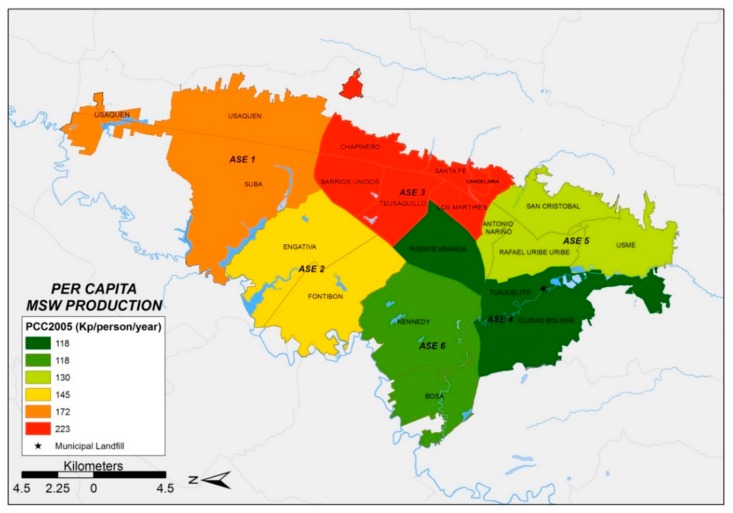
Municipal solid waste (MSW) generation per capita of each exclusive service area (ASE) (2005).

**Figure 2 ijerph-17-01196-f002:**
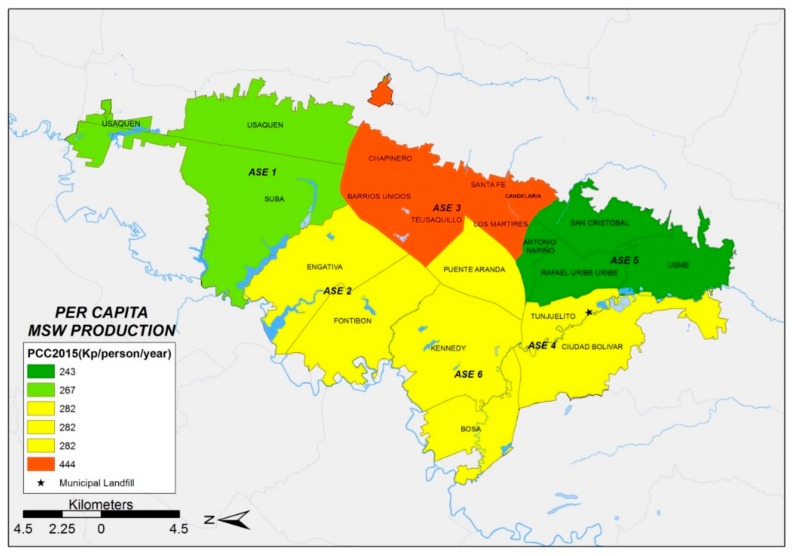
Municipal solid waste generation per capita of each ASE (2015).

**Figure 3 ijerph-17-01196-f003:**
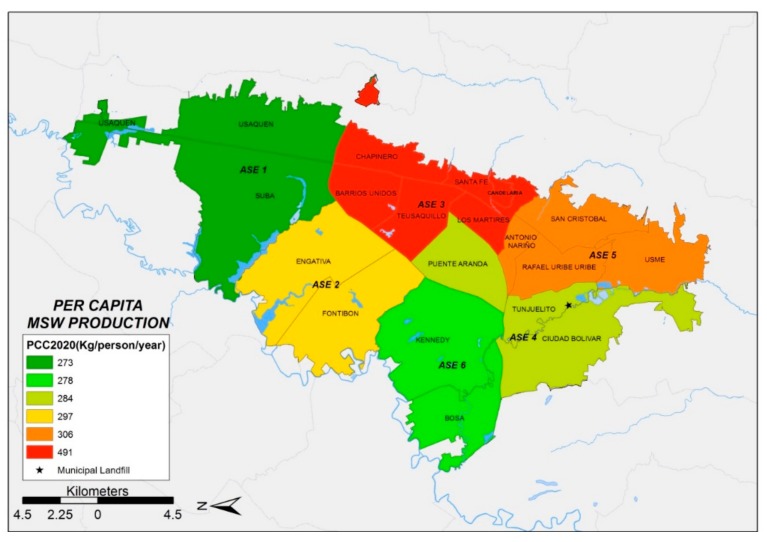
Municipal solid waste generation per capita of each ASE (2020).

**Figure 4 ijerph-17-01196-f004:**
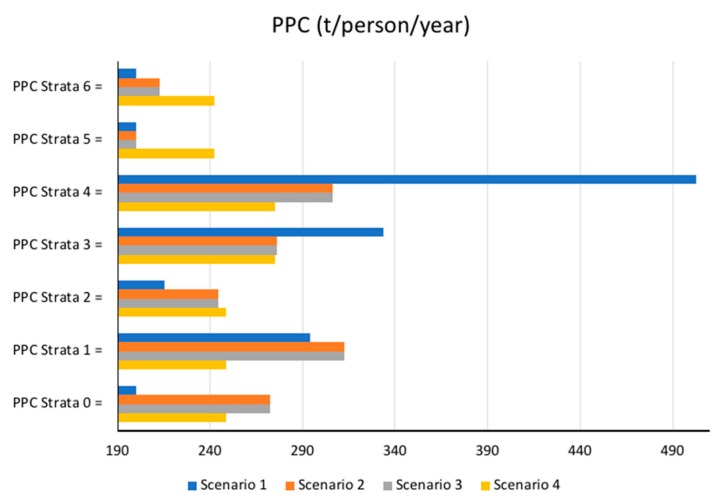
Solid waste generation per capita of stratum i (PPCstr,i) for Scenarios 1 to 4.

**Table 1 ijerph-17-01196-t001:** Household distribution by the socioeconomic stratum of each district for 2011 [19,20].

ASE	District	Number Households	Str 0 (%)	Str 1 (%)	Str 2 (%)	Str 3 (%)	Str 4 (%)	Str 5 (%)	Str 6 (%)
5	Antonio Nariño	27,774	1.79	0.00	4.49	93.72	0.00	0.00	0.00
3	Barrios Unidos	57,196	0.16	0.00	0.00	54.60	41.89	3.35	0.00
6	Bosa	133,097	4.51	4.83	87.18	3.48	0.00	0.00	0.00
3	Chapinero	55,919	0.21	3.38	9.41	6.30	33.18	10.45	37.07
4	Ciudad Bolívar	152,266	1.21	60.02	34.56	4.22	0.00	0.00	0.00
2	Engativá	232,205	1.04	0.74	23.58	70.63	4.00	0.00	0.00
2	Fontibón	116,233	1.94	0.00	18.76	44.66	33.75	0.88	0.00
6	Kennedy	269,028	0.75	0.58	49.37	46.80	2.50	0.00	0.00
3	La Candelaria	7857	0.71	0.45	53.30	45.54	0.00	0.00	0.00
3	Los Mártires	27,497	0.18	0.00	8.79	83.29	0.00	0.00	0.00
4	Puente Aranda	70,682	1.05	0.00	0.32	98.62	0.00	0.00	0.00
5	Rafael Uribe	104,433	1.12	8.53	47.72	42.63	0.00	0.00	0.00
5	San Cristóbal	112,721	0.39	7.74	77.40	14.47	0.00	0.00	0.01
3	Santa Fe	36,163	1.51	7.49	55.47	25.13	9.39	0.49	0.51
1	Suba	288,568	1.61	0.21	30.56	33.59	18.78	13.71	1.54
5	Sumapaz	1743	0.00	54.91	28.06	9.70	3.61	1.61	2.12
3	Teusaquillo	57,972	0.16	0.00	0.00	14.12	80.71	5.02	0.00
4	Tunjuelito	49,168	0.49	0.00	56.48	43.04	0.00	0.00	0.00
1	Usaquén	162,641	0.94	4.36	6.22	26.16	30.47	13.72	18.14
5	Usme	99,114	2.20	47.10	50.69	0.01	0.00	0.00	0.00

**Table 2 ijerph-17-01196-t002:** Solid waste generation of each ASE for the period 2005–2020.

Year	ASE	Solid Waste Generation (t/year)	Population	PPC (kg/person/year)
2005	ASE 1	235,201.40	1,363,504	173
	ASE 2	158,763.73	1,091,877	145
	ASE 3	160,571.69	718,797	223
	ASE 4	121,561.06	1,027,293	118
	ASE 5	155,205.83	1,192,633	130
	ASE 6	170,779.33	1,440,060	119
2015	ASE 1	447,188.72	1,668,802	268
	ASE 2	354,295.06	1,255,208	282
	ASE 3	338,561.67	762,829	444
	ASE 4	323,844.30	1,146,385	282
	ASE 5	321,634.07	1,322,797	243
	ASE 6	484,009.26	1,716,302	282
2020	ASE 1	507,542.87	1,858,528	273
	ASE 2	396,484.94	1,337,120	296
	ASE 3	366,090.73	746,291	490
	ASE 4	332,914.65	1,171,220	284
	ASE 5	362,623.87	1,186,754	305
	ASE 6	576,760.04	2,073,050	278

**Table 3 ijerph-17-01196-t003:** Population distribution by economic stratum of each ASE (2015. The population is specified in this table by economic stratum in each of the solid waste collection areas.

ASE	%pop	Population	Pop Str 0	Pop Str 1	Pop Str 2	Pop Str 3	Pop Str 4	Pop Str 5	Pop Str 6
ASE 1	22%	1,723,691	24,582	23,246	411,639	543,835	378,983	236,359	105,047
ASE 2	16%	1,276,634	16,910	6462	281,611	796,983	171,117	3551	0
ASE 3	9%	724,902	2702	11,623	85,831	284,921	262,626	29,646	47,553
ASE 4	14%	1,134,511	12,002	431,964	356,491	334,054	0	0	0
ASE 5	15%	1,196,612	14,881	219,654	651,424	310,613	0	0	40
ASE 6	24%	1,896,355	40,883	41,133	1,204,318	580,338	29,683	0	0
TOTAL			111,960	734,082	2,991,314	2,850,744	842,409	269,556	152,640
Population %			1.4%	9.2%	37.6%	35.8%	10.6%	3.4%	1.9%

**Table 4 ijerph-17-01196-t004:** Objective function minimization calculations for Scenarios 1 to 4.

Scenario	ASE	Estimated Generation (t/Year)	Real Generation (t/Year)	Difference (t/Year)	Error	Absolute Difference (t/Year)	Absolute Error	Objective Function (t/Year)
1	ASE 1	540,544	450,453	90,091	20%	90,091	20.0%	38,034
	ASE 2	418,445	348,704	69,741	20%	69,741	20.0%	
	ASE 3	264,944	324,087	−59,143	−18%	59,143	18.2%	
	ASE 4	317,633	317,633	0	0%	0	0.0%	
	ASE 5	311,384	324,146	−12,762	−4%	12,762	3.9%	
	ASE 6	487,897	487,900	−3	0%	3	0.0%	
						Σ = 231,737		
2	ASE 1	450,454	450,453	1	0%	1	0.0%	12,238
	ASE 2	348,612	348,704	−92	0%	92	0.0%	
	ASE 3	200,484	324,087	−123,603	−38%	123,603	38.1%	
	ASE 4	317,634	317,633	1	0%	1	0.0%	
	ASE 5	317,789	324,146	−6357	−2%	6357	2.0%	
	ASE 6	487,900	487,900	0	0%	0	0.0%	
						Σ = 130,054		
3	ASE 1	450,452	450,453	−1	0%	1	0.0%	1147
	ASE 2	348,713	348,704	9	0%	9	0.0%	
	ASE 4	317,632	317,633	−1	0%	1	0.0%	
	ASE 5	317,230	324,146	−6916	−2%	6919	2.1%	
	ASE 6	487,900	487,900	0	0%	0	0.0%	
						Σ = 6927		
4	ASE 1	450,836	450,453	383	0%	383	0.1%	8285
	ASE 2	342,983	348,704	−5721	−2%	5721	1.6%	
	ASE 4	291,084	317,633	−26,549	−8%	26,549	8.4%	
	ASE 5	305,928	324,146	−18,218	−6%	18,218	5.6%	
	ASE 6	487,900	487,900	0	0%	0	0.0%	
						Σ = 50,871		

**Table 5 ijerph-17-01196-t005:** Solid waste generation per capita of stratum i (PPCstr,i) for Scenarios 1 to 4.

Scenario	Units	PPCstr,i						
		PPC Str 0	PPC Str 1	PPC Str 2	PPC Str 3	PPC Str 4	PPC Str 5	PPC Str 6
1	t/person/year	200	294	215	334	503	200	200
	kg/person/day	0.55	0.81	0.59	0.91	1.38	0.55	0.55
2	t/person/year	272	312	245	276	306	200	213
	kg/person/day	0.75	0.86	0.67	0.76	0.84	0.55	0.58
3	t/person/year	288	305	239	291	246	245	275
	kg/person/day	0.79	**0.84**	0.65	0.80	0.67	0.67	0.75
4	t/person/year	249	**249**	249	275	275	242	242
	kg/person/day	0.68	**0.68**	0.68	0.75	0.75	0.66	0.66

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
