# Peer review of "Analytical Methodology for the Identification of Critical Zones on the Generation of Solid Waste in Large Urban Areas"

_ijerph, 2020, doi:10.3390/ijerph17041196_

Round 1
Reviewer 1 Report
The authors attempt to provide an analytical framework for municipal solid waste, however the overall research study is unclear and rewrite is necessary for resubmission.
The authors have not shown clear focus, as well as clear research questions for this study. The authors have not given relevant literature reviews for the research topic. The authors did not provide any concrete analytical framework for this research.Author Response
Please see the attachment

Reviewer 2 Report
This paper presents an interesting GIS analysis methodology to identify the solid waste generation in Bogota.
Overall, the content is concisely written. Below are some minor comments:
1. Page 3. Table 1. Please add a short description of the meaning of strata levels, e.g. the higher the wealthier or otherwise, as most reader will unlikely to be familiar with the classification.
2. Page 4. Meaning of PPCstr i and the objective function is not clearly explained. What does it imply?
3. Figure 1. Please increase the resolution of the figure. The wording on the map cannot be read clearly.
4. Page 8. Please elaborate what is the four scenarios represent. We can understand the calculation, but what is the meaning of considering the four scenarios?
Reviewer 3 Report
Excellent manuscript, recommended for publication
Round 2
Reviewer 1 Report
Its much more improved after read the revised manuscript. And the reviewer accept the exiting version of manuscript without further revision.
Author Response
Thank you very much for your comments